# Circulating Osteopontin Levels and Outcomes in Patients Hospitalized for COVID-19

**DOI:** 10.3390/jcm10173907

**Published:** 2021-08-30

**Authors:** Salim S. Hayek, Christoph Roderburg, Pennelope Blakely, Christopher Launius, Jesper Eugen-Olsen, Frank Tacke, Sofia Ktena, Verena Keitel, Mark Luedde, Evangelos J. Giamarellos-Bourboulis, Tom Luedde, Sven H. Loosen

**Affiliations:** 1Division of Cardiology, Department of Internal Medicine, University of Michigan, Ann Arbor, MI 48109, USA; pblakely@med.umich.edu (P.B.); claunius@med.umich.edu (C.L.); 2Clinic for Gastroenterology, Hepatology and Infectious Diseases, Medical Faculty, University Hospital Düsseldorf, 40225 Düsseldorf, Germany; christoph.roderburg@med.uni-duesseldorf.de (C.R.); Verena.Keitel@med.uni-duesseldorf.de (V.K.); sven.loosen@med.uni-duesseldorf.de (S.H.L.); 3Department of Clinical Research, Copenhagen University Hospital Amager and Hvidovre, 2650 Hvidovre, Denmark; jespereugenolsen@gmail.com; 4Department of Hepatology and Gastroenterology, Charité-Universitätsmedizin Berlin, 10117 Berlin, Germany; frank.tacke@charite.de; 54th Department of Internal Medicine, National and Kapodistrian University of Athens, Medial School, 12462 Athens, Greece; sktena@med.uoa.gr (S.K.); egiamarel@med.uoa.gr (E.J.G.-B.); 6KGP Bremerhaven, Postbrookstraße 103, 27574 Bremerhaven, Germany; mark.luedde@web.de

**Keywords:** OPN, CRP, procalcitonin, PCT, coronavirus disease 2019, SARS-CoV-2, outcomes, risk prediction, death, mechanical ventilation, renal replacement therapy

## Abstract

Background: Severe coronavirus disease 2019 (COVID-19) is the result of a hyper-inflammatory reaction to the severe acute respiratory syndrome coronavirus 2. The biomarkers of inflammation have been used to risk-stratify patients with COVID-19. Osteopontin (OPN) is an integrin-binding glyco-phosphoprotein involved in the modulation of leukocyte activation; its levels are associated with worse outcomes in patients with sepsis. Whether OPN levels predict outcomes in COVID-19 is unknown. Methods: We measured OPN levels in serum of 341 hospitalized COVID-19 patients collected within 48 h from admission. We characterized the determinants of OPN levels and examined their association with in-hospital outcomes; notably death, need for mechanical ventilation, and need for renal replacement therapy (RRT) and as a composite outcome. The risk discrimination ability of OPN was compared with other inflammatory biomarkers. Results: Patients with COVID-19 (mean age 60, 61.9% male, 27.0% blacks) had significantly higher levels of serum OPN compared to healthy volunteers (96.63 vs. 16.56 ng/mL, *p* < 0.001). Overall, 104 patients required mechanical ventilation, 35 needed dialysis, and 53 died during their hospitalization. In multivariable analyses, OPN levels ≥140.66 ng/mL (third tertile) were associated with a 3.5 × (95%CI 1.44–8.27) increase in the odds of death, and 4.9 × (95%CI 2.48–9.80) increase in the odds of requiring mechanical ventilation. There was no association between OPN and need for RRT. Finally, OPN levels in the upper tertile turned out as an independent prognostic factor of event-free survival with respect to the composite endpoint. Conclusion: Higher OPN levels are associated with increased odds of death and mechanical ventilation in patients with COVID-19, however, their utility in triage is questionable.

## 1. Introduction

COVID-19 represents a hyperinflammatory condition characterized by an excessive immune activation and an overshooting release of cytokines often leading to multi-organ failure [1,2,3]. However, the individual course of COVID-19 is heterogeneous and early identification of patients at risk for a severe course of disease is often not possible. Inflammatory biomarkers such as C-reactive protein (CRP), interleukin-6 (IL-6), and procalcitonin, often used in critically ill patients, have been proposed as risk stratification tools in the context of COVID-19 [3,4,5]. Nevertheless, distinguishing reliably between patients who can be safely discharged and those who need a higher level of care remains challenging.

Osteopontin (OPN) is a 44 to 66 kDa integrin-binding extracellular matrix glyco-phosphoprotein involved in inflammatory response [6]. OPN modulates leucocyte differentiation, migration, and activation as well as the secretion of various cytokines during acute and chronic inflammation. Elevated levels of circulating OPN have been reported in patients with systemic inflammation such as sepsis [7,8] and are associated with increased mortality rates in these patients [9]. Alveolar macrophages strongly express OPN in patients with non-COVID-19 related acute respiratory distress syndrome (ARDS) (7), and preliminary results from smaller cohorts suggest a predictive role of OPN in the context of COVID-19 [10].

We hypothesize that circulating levels of OPN are elevated in patients with COVID-19 and associated with worse in-hospital outcomes. To test our hypothesis, we leveraged the International Study of Inflammation in COVID-19 (ISIC) to measure serum OPN levels in a subset of its patients and examined their association with in-hospital death and need for mechanical ventilation or renal replacement therapy.

## 2. Patients and Methods

### 2.1. Study Design

We measured OPN levels in serum samples collected within 48 h of admission of patients hospitalized for COVID-19 enrolled in ISIC. The International Study of Inflammation in COVID-19 is an ongoing multi-center observational study that aims to characterize the role of inflammation in COVID-19 through the measurement of various blood-based biomarkers [11]. Patients were enrolled in ISIC if they were adults ≥18 years hospitalized primarily for COVID-19, had a SARS-CoV-2 infection confirmed by reverse transcriptase polymerase chain reaction test of nasopharyngeal or oropharyngeal samples, and had a biological sample collected (plasma or serum) within 48 h of hospitalization. Clinical data collected included symptoms at presentation, past medical history, home medications, inpatient medical therapy, biomarker levels, and hospitalization course and outcomes. The data was extracted through manual chart review by at least two reviewers per site and entered in an online data repository managed using REDCap electronic data capture tools hosted at the University of Michigan. Institutional review board approval and consent procedures were obtained separately at each site according to local institutional policies (approval reference: HUM00178971 (Michigan), IS 021-20 (Athens), and #5350 (Duesseldorf)). 

For the purpose of this study, 341 participants of ISIC with available serum samples from the following centers were included: University of Michigan in Ann Arbor, MI USA (*n* = 217); Attikon University Hospital in Athens, Greece (*n* = 89); University Hospital of Dusseldorf, Germany (*n* = 28); Aachen University Hospital, Germany (*n* = 7). Participants in this ancillary study were randomly selected from a subset of ISIC patients who were hospitalized during the first COVID-19 surge from 1 February to 1 May 2020 after excluding those mechanically ventilated within 48 h of presentation. 

### 2.2. Healthy Cohort

We measured OPN in 40 healthy blood donors without preexisting infectious, cardio-pulmonal, or malignant disease in order to compare OPN levels to that of patients with COVID-19. 

### 2.3. Outcomes and Measures

The main outcomes examined were in-hospital death, need for mechanical ventilation, and need for renal replacement therapy (RRT), both as a composite outcome and individually. Circulating OPN concentrations were measured using a commercially available enzyme-linked immunosorbent assay (ELISA) that detects both full-length and cleaved human OPN according to the manufacturer’s instructions (#BMS2066, Thermo Fischer Scientific, Waltham, MA, USA). C-reactive protein, ferritin, D-dimer and procalcitonin levels were measured at the request of the clinical team by the central laboratory of the respective institution of enrollment.

### 2.4. Statistical Analysis

We first report demographic and clinical characteristics stratified by OPN tertiles (<55.81 ng/mL [*n* = 113], 55.81–140.65 ng/mL [*n* = 114], and ≥140.66 ng/mL [*n* = 114]), and present continuous variables as means (±standard deviation [SD]) or as median (25th to 75th interquartile range) for normally and non-normally distributed data, respectively. Categorical variables are presented as proportions (%). We used ANOVA, Kruskal–Wallis, and Chi-square tests to compare characteristics and outcomes between OPN tertiles. Spearman-rank was used to assess correlations between OPN and C-reactive protein, D-dimer, ferritin and procalcitonin biomarkers. 

To identify determinants of OPN levels, we used multivariable linear regression with OPN levels as the dependent variable and included age, gender, race, body-mass index, smoking, admission creatinine-derived estimated glomerular filtration rate (eGFR), diabetes mellitus, hypertension, coronary artery disease, and heart failure in the model. 

To evaluate the association between OPN and outcomes, we log-transformed OPN levels (base 2, interpreted as per 100% increase) and used binary logistic regression modeling including the following clinical characteristics: age, sex, race, body-mass index, smoking, eGFR on admission, diabetes mellitus, hypertension, congestive heart failure. We report adjusted odds ratios for each OPN tertile, with the lowest tertile serving as the reference group. We also performed a time-to-event analysis using Cox regression modeling using the composite outcome of death, mechanical ventilation and need for RRT. 

Lastly, we compared the ability of OPN to that of C-reactive protein in discriminating risk of the composite outcome by computing areas under the curves (AUC) and comparing the AUCs using the DeLong’s test. We provided OPN’s test characteristics (sensitivity, specificity, positive predictive value and negative predictive value) for predicting the composite outcome using the highest tertile as a cut-off. We considered two-tailed *p*-values ≤ 0.05 as statistically significant, and performed the analyses using SPSS 24 (IBM, New York, NY, USA).

## 3. Results

### 3.1. Characteristics of Study Cohort

A total of 341 COVID-19 patients from ISIC met the inclusion criteria for this study. The mean age of the study population was 61 years (SD: 16). Of the total number of patients, 61.9% were male and 27.0% were black. The mean BMI of the study cohort was 30.79 kg/m^2^ (SD: 7.58). The composite endpoint of need for mechanical ventilation (*n* = 104) or need for renal replacement therapy (*n* = 35) or death (*n* = 53) was met by a total of 192 patients (56.3%). Table 1 provides an overview of the study cohort.

### 3.2. Circulating Levels of OPN Are Elevated in Patients Hospitalized for COVID-19

We first compared circulating levels of OPN between healthy controls (*n* = 40) and COVID-19 patients. OPN levels measured within 48 h of hospital admission were significantly higher in COVID-19 patients compared to healthy controls (median OPN level: 16.56 vs. 80.42 ng/mL, *p* < 0.001, Figure 1). To identify the potential drivers of elevated OPN levels in patients with COVID-19, we performed linear regression models that revealed a positive association between OPN serum levels and patients’ race as well as glomerular filtration rate (Table 2). Moreover, we observed significant correlations between circulating OPN levels and other routine biomarkers: CRP (r_S_: 0.396, *p* < 0.001), procalcitonin (PCT, r_S_: 0.396, *p* < 0.001), ferritin (r_S_: 0.310, *p* < 0.001), and d-dimers (r_S_: 0.193, *p* < 0.001).

### 3.3. Elevated Levels of OPN Are Associated with an Increased Risk of Death and Need of Mechanical Ventilation

We next examined the ability of OPN measurements to predict the clinical course of COVID-19 patients. Patients in the first tertile of OPN levels had a mortality rate of 10.6%, compared to 27.2% for patients in the 3rd OPN tertile (Table 3). In multivariable binary logistic regression analysis, OPN levels within the third tertile were an independent predictor of death (OR: 3.450, 95%CI: 1.439–8.273, *p* = 0.006, Table 4). Other characteristics associated with mortality included age, race, smoking status, and preexisting diabetes mellitus (Table 4). 

As many patients with an unfavorable clinical course require mechanical ventilation (MV) or renal replacement therapy (RRT) at some point during treatment, we investigated the predictive function of OPN concentrations upon hospital admission for these clinical endpoints. We observed a significant increase in MV rates from the lowest (15.9%) to the highest OPN tertile (50.1%, Table 3). OPN levels within the 3rd tertile further turned out as independent predictor for a need of MV in logistic regression analysis (OR: 4.931, 95%CI: 2.480–9.804, *p* < 0.001, Table 4). Besides OPN, the patients’ BMI and preexisting diabetes mellitus were independent predictors for a need of MV (Table 4). In contrast OPN measurements did not allow predicting the need of RRT in our cohort of COVID-19 patients (Table 3 and Table 4).

### 3.4. Circulating OPN as a Predictor of Severe Clinical Course of Disease

The prediction of a combined composite endpoint (in-hospital death and/or need for MV and/or need for RRT) might represent a clinically more relevant tool to identify patients at high risk for complicated clinical course. We noted a stepwise increase in the incidence of the composite endpoint amongst patients within higher OPN tertiles (Table 3). In detail, patients in the second and third OPN tertiles had a 1.19 × and 3.23 × increase in the odds of the composite outcome, respectively, compared to the first tertile. ROC curve analysis revealed an area under the curve (AUC) value of 0.677 for OPN concentrations upon admission regarding the composite endpoint, which was only slightly inferior to serum CRP values (AUC_CRP_: 0.725, Figure 2). The combination of OPN and CRP serum levels had the highest accuracy for the prediction of the composite endpoint (AUC_OPN/CRP_: 0.728, Figure 2)

We finally performed time-to-event analyses for the composite endpoint. Kaplan–Meier curve analysis revealed a significantly better event-free survival for COVID-19 patients who presented with initial OPN levels within the lower tertile (Figure 3). Patients with circulating OPN levels in the upper tertile had a median event-free survival of only 6 days (Figure 3). The prognostic relevance of OPN upon hospital admission was further corroborated in a multivariate Cox-regression model. Here, OPN levels in the upper tertile turned out as an independent prognostic factor regarding event-free survival (HR: 1.834, 95% CI: 1.071–3.139, *p* = 0.027, Table 5). In addition, the patients’ BMI, eGFR, preexisting diabetes mellitus as well as circulating CRP levels were of prognostic relevance with respect to event-free survival (Table 5).

## 4. Discussion

The COVID-19 pandemic poses an unprecedented challenge to even well-resourced healthcare systems. The unexpectedly high number of patients requiring hospitalization or even intensive care therapy resulted in overtaxing of medical systems and the inability to treat all patients optimally. The clinical course of SARS-CoV-2 infections is highly variable and can be only poorly predicted by classical clinical markers. However, an early risk stratification of COVID-19 patients upon hospital admission is essential not only to provide high-risk patients with the best possible medical treatment but also to optimally allocate the available medical resources and to prevent a health care system overload. In the present study, we report that circulating levels of OPN, an integrin-binding extracellular matrix glycol-phosphoprotein, are elevated in patients with COVID-19 upon hospital admission. We show that concentrations of OPN directly reflect disease severity and represent an independent risk factor for a more severe clinical course. As such, patients with high OPN levels at hospital admission were at higher need for mechanical ventilation and had a significantly impaired survival. Thus, this work supports the development of novel markers to aid clinical decision making and generate hypotheses about potential COVID-19 therapeutic targets.

Several clinical, medical, societal, and economic factors have recently been identified as major determinants of severe clinical courses and mortality in COVID-19 patients [12,13]. To name few, older age, high SOFA score, and d-dimer greater than 1 μg/mL were identified as negative risk factors associated with a poor prognosis at an early stage of disease [12,13]. In addition, interleukin-6 (IL-6) was suggested as a biomarker for more severe courses of COVID-19 [14], and tocilizumab, a humanized monoclonal antibody targeting the IL-6 receptor, has been evaluated as a potential therapeutic option for COVID-19 patients and was shown to improve outcomes, e.g., in critically ill patients with COVID-19 receiving organ support in ICUs [15,16]. Along with elevated OPN, in our analysis comprising 341 patients treated for COVID-19 in 4 tertiary care centers in 4 Western countries, older age, black race, current or former smoker status, and diabetes mellitus turned out as independent factors associated with patients’ mortality. When extended to a broader combined end-point including death, need for mechanical ventilation or need for renal replacement therapy OPN, eGFR, elevated BMI and diabetes mellitus predicted unfavorable disease courses. Importantly, in all these analyses, OPN represented the strongest parameter for the prediction of severe disease courses in COVID-19 patients. 

OPN is a multifaceted molecule involved in the inflammatory response: it modulates the recruitment of monocytes-macrophages and regulates cytokine production in macrophages, dendritic cells, and T-cells. OPN has been classified as T-helper 1 cytokine and thus believed to exacerbate inflammation in several chronic inflammatory diseases, including lung diseases. In line to our findings, the working groups of Varim et al. described elevated levels of OPN in patients with COVID-19 and suggested a relationship between serum osteopontin levels and the disease severity [17]. OPN is reportedly associated with a broad spectrum of inflammatory and malignant diseases [7,18,19]. In line, along with elevated concentrations of OPN in COVID-19 patients, we found a strong correlation of OPN levels with different marker for systemic inflammation (CRP, lactate dehydrogenase D-dimer, ferritin, and procalcitonin). In a large and well characterized cohort of critically ill patients, we recently demonstrated a clear association between elevated OPN levels and patient’s prognosis during ICU treatment, confirming the results of this study. More specifically, increased OPN-levels were described in patients with interstitial [20] and eosinophilic pneumonias [21]. In another analyses featuring patients hospitalized for respiratory failure, OPN-plasma levels were found to be increased compared to healthy controls, whereas no difference was found between SARS-CoV-2 positive and SARS-CoV-2 negative patients [10]. These findings suggest that OPN cannot be used as a biomarker supporting the diagnosis of SARS-CoV-2 infection but rather as a marker for stratifying patients according to their clinical course. To better understand factors determining OPN concentrations in COVID-19 we performed linear regression analyses including several patients´ and disease related factors. However, only black race, eGFR and CRP (*p* = 0.055) turned out as factors associated with circulating OPN levels. On the other hand, it was recently reported that specific antibodies directed against OPN are produced in the context of diseases associated with systemic inflammation (e.g., multiple sclerosis and experimental autoimmune encephalomyelitis) and might neutralize the effect of elevated OPN levels [22]. Thus, in addition to OPN concentrations themselves, the specific effects of OPN represent a complex and insufficiently understood, demanding further research e.g., in larger prospective trials or animal models. 

The present study has several strengths. First, the multinational study design at four study sites in the USA and central Europe. Second, the high degree of standardization in sample collection and analyses argues for the high reliability of our results. However, we do acknowledge some limitations, with the high number of patients with incomplete sets of biomarker measures being the most important one. This is due to the fact that first routine clinical markers were performed following standards of the local investigators and second the availability of OPN measurements was limited by the ability of local centers to collect samples within 2 days by the study team and often small residual volumes. However, we want to emphasize that the “missingness” of these data appeared to be random, limiting the bias introduced by missing data. Furthermore, there was an important difference in enrollment across centers, however, sensitivity analyses did not show differences in the association between biomarker and outcomes according to center. 

In summary, our data unraveled a previously unrecognized role of circulating OPN as a novel predictive and prognostic biomarker that enables the identification of COVID-19 patients at risk of experiencing a more severe course of disease. This easily accessible risk stratification tool might help to guide treatment decisions and facilitate an optimal distribution of medical resources during this ongoing pandemic when implemented in a panel of other of routinely accessed markers systemic inflammation and/or infection, which are widely available and analyzed in almost all patients suspected for COVID-19.

## Figures and Tables

**Figure 1 jcm-10-03907-f001:**
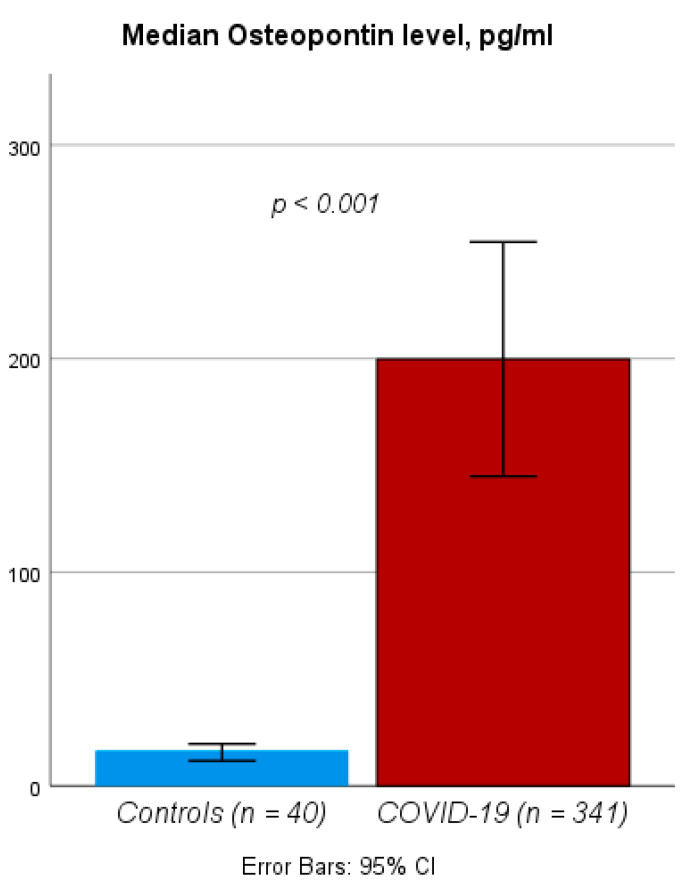
OPN levels in COVID-19 patients and healthy controls. Circulating OPN levels are significantly elevated in COVID-19 upon admission to the hospital compared to healthy controls.

**Figure 2 jcm-10-03907-f002:**
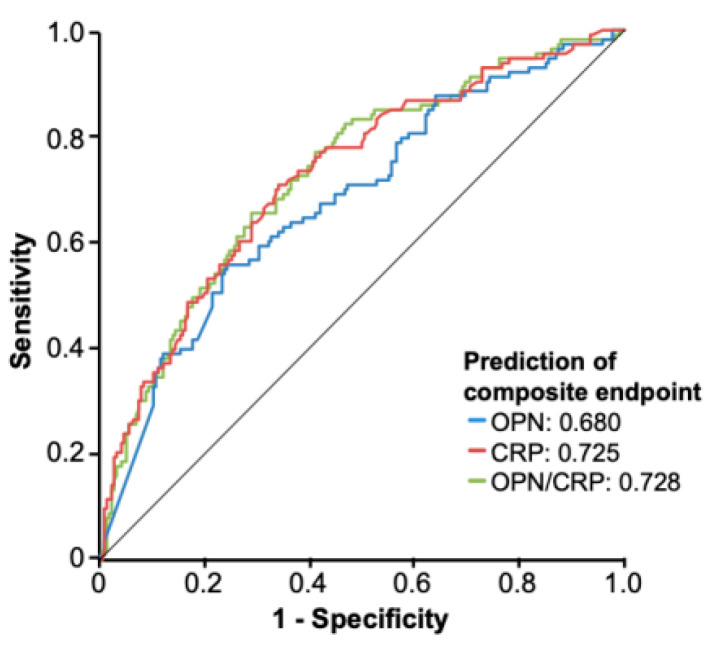
ROC curve analysis for the prediction of the composite endpoint. The combination of OPN and CRP serum levels has the highest accuracy for the prediction of the composite endpoint (either death, need for mechanical ventilation or need for renal replacement therapy).

**Figure 3 jcm-10-03907-f003:**
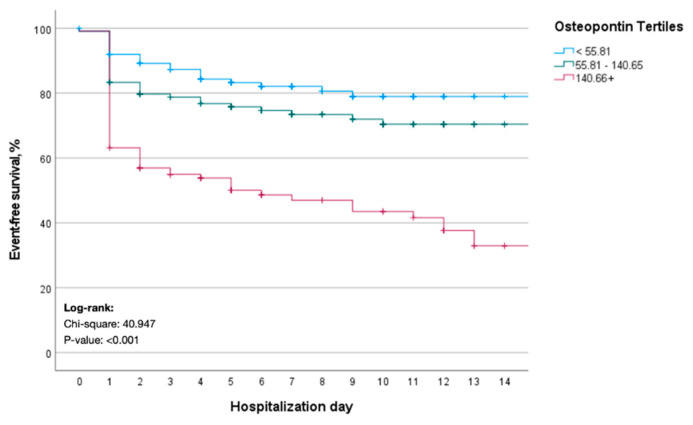
Kaplan-Meier curve estimates for the composite endpoint. COVID-19 patients with initial OPN levels in the lower tertile have a significantly better event-free survival regarding the composite endpoint (either death, need for mechanical ventilation or need for renal replacement therapy).

**Table 1 jcm-10-03907-t001:** Clinical characteristics of the study cohort.

	All (*n* = 341)
Clinical characteristics	
Age in years (mean, SD)	61 (16)
<45, *n* (%)	52 (15.2%)
45–64, *n* (%)	150 (44.0%)
65–79, *n* (%)	95 (27.9%)
≥80, *n* (%)	44 (12.9%)
Male, *n* (%)	211 (61.9%)
White race, *n* (%)	224 (65.7%)
Black race, *n* (%)	92 (27.0%)
BMI kg/m^2^, mean (SD)	30.79 (7.58)
Current or former smoker, *n* (%)	112 (32.8%)
Diabetes mellitus, *n* (%)	116 (34.0%)
Hypertension, *n* (%)	206 (60.4%)
Coronary artery disease, *n* (%)	47 (13.8%)
Congestive heart failure, *n* (%)	47 (13.8)
Laboratory parameters	
Admission eGFR, mL/min/1.73 m^2^, mean (SD)	70.61 (33.02)
White cell count, 10^3^ ×, mean (SD)	7.6 (4.2)
Absolute neutrophil count, 10^3^ ×, mean (SD)	5.8 (3.6)
Absolute lymphocyte count, 10^3^ ×, mean (SD)	1.1 (1.7)
Aspartate aminotransferase, IU/L, mean (SD)	56.85 (49.53)
Alanine aminotransferase, IU/L, mean (SD)	41.6 (51.3)
Bilirubin, mg/dl, mean (SD)	0.65 (0.39)
Alkaline phosphatase, IU/L, mean (SD)	81 (46)
C-reactive protein, mg/dl, median (IQR)	8.3 (3.6–17.3)
D-dimer, FEU mg/l, median (IQR)	1.085 (0.57–2.18)
Ferritin, ng/mL, median (IQR)	613.15 (268.6–1321.7)
Procalcitonin, ng/mL, median (IQR)	0.2 (0.1–0.77)
OPN, ng/mL, median (IQR)	80.42 (43.44–164.0)
Outcomes	
Need for MV, *n* (%)	104 (30.5%)
Need for RRT, *n* (%)	35 (10.3%)
Death, *n* (%)	53 (15.5%)

Abbreviations: BMI: body mass index; eGFR: estimated glomerular filtration rate; IQR: interquartile range; IU: international units; SD: standard deviation; OPN: osteopontin, MV: mechanical ventilation; RRT: renal replacement therapy.

**Table 2 jcm-10-03907-t002:** Determinants of OPN (linear regression model).

Parameter	β (95%CI)	*p*-Value
Age	−6.49 (−38.03–25.05)	0.260
Male sex	20.13 (2.95–37.32)	0.690
Black race	−1.39 (−3.58–0.80)	0.022
BMI	−0.94 (−1.53–0.36)	0.210
eGFR	3.22 (−0.51–6.94)	0.002
WBC	0.63 (−0.02–1.27)	0.090
CRP	28.66 (−6.66–63.99)	0.060
Diabetes mellitus	−3.35 (−41.45–34.76)	0.110
Hypertension	−22.22 (−70.73–26.30)	0.860
Coronary artery disease	38.66 (−9.98–87.29)	0.370
Congestive heart failure	−6.49 (−38.03–25.05)	0.120

BMI: body mass index, CI: confident interval eGFR: estimated glomerular filtration rate, WBC: white blood cells, CRP: C-reactive protein.

**Table 3 jcm-10-03907-t003:** Clinical characteristics and inflammatory markers stratified by OPN tertiles.

	1st Tertile (OPN < 55.81 ng/mL, *n* = 113)	2nd Tertile (OPN 55.81–140.65 ng/mL, *n* = 114)	3rd Tertile (OPN ≥ 140.66 ng/mk, *n* = 114)	*p*-Value
Clinical characteristics				
Age in years (mean, SD)	58 (16)	62 (15)	61 (16)	0.120
<45, *n* (%)	19 (16.8%)	16 (14.0%)	17 (14.9%)	0.380
45–64, *n* (%)	57 (50.4%)	46 (40.4%)	47 (41.2%)	0.240
65–79, *n* (%)	28 (24.8%)	36 (31.6%)	31 (27.2%)	0.510
≥80, *n* (%)	9 (8.0%)	16 (14.0%)	19 (16.7%)	0.130
Male, *n* (%)	64 (56.6%)	74 (64.9%)	73 (64.0%)	0.370
White race, *n* (%)	86 (76.1%)	78 (68.4%)	60 (52.6%)	0.022
Black race, *n* (%)	20 (17.7%)	29 (25.4%)	43 (37.7%)	0.022
BMI kg/m^2^, mean (SD)	30.95 (7.68)	30.7 (7.67)	30.75 (7.46)	0.970
Current or former smoker, *n* (%)	31 (27.4%)	40 (35.1%)	41 (36.0%)	0.175
Diabetes mellitus, *n* (%)	28 (24.8%)	43 (37.7%)	45 (39.5%)	0.039
Hypertension, *n* (%)	59 (52.2%)	71 (62.3%)	76 (66.7%)	0.074
Coronary artery disease, *n* (%)	14 (12.4%)	16 (14.0%)	17 (14.9%)	0.850
Congestive heart failure, *n* (%)	11 (9.7%)	14 (12.3%)	22 (19.3%)	0.096
Laboratory parameters				
Admission eGFR, mL/min/1.73 m^2^, mean (SD)	76.77 (31.5)	72.85 (30.83)	62.26 (35.11)	0.003
White cell count, 10^3^×, mean (SD)	6.9 (3.4)	7.4 (4.1)	8.6 (4.9)	0.005
Absolute neutrophil count, 10^3^×, mean (SD)	5.1 (3.0)	5.6 (3.7)	6.8 (3.9)	0.001
Absolute lymphocyte count, 10^3^×, mean (SD)	1.1 (0.5)	1.1 (0.5)	1.1 (2.8)	0.970
Aspartate aminotransferase, IU/L, mean (SD)	50.29 (61.74)	54.47 (35.21)	65.67 (46.96)	0.054
Alanine aminotransferase, IU/L, mean (SD)	43.8 (78.9)	40.5 (28)	40.5 (30.1)	0.855
Bilirubin, mg/dl, mean (SD)	0.61 (0.35)	40.5 (0.39)	0.62 (0.42)	0.079
Alkaline phosphatase, IU/L, mean (SD)	75 (43)	85 (49)	84 (46)	0.266
C-reactive protein, mg/dl, median (IQR)	4.6 (2.1–11.0)	7.86 (3.9–16.2)	14.35 (7.28–23.1)	<0.001
D-dimer, FEU mg/l, median (IQR)	0.7 (0.48–2.025)	0.965 (0.56–1.985)	1.33 (0.69–2.97)	0.013
Ferritin, ng/mL, median (IQR)	356.65 (198–1094.5)	625.5 (316.3–1260.3)	904.9 (353.9–1530.6)	<0.001
Procalcitonin, ng/mL, median (IQR)	0.135 (0.07–0.305)	0.19 (0.1–0.42)	0.54 (0.15–2.07)	<0.001
OPN, ng/mL, median (IQR)	28.17 (19.87–43.17)	80.23 (66.91–105.61)	372.53 (164–400)	<0.001
Outcomes				
Need for MV, *n* (%)	18 (15.9%)	29 (25.4%)	57 (50.0%)	<0.001
Need for RRT, *n* (%)	10 (8.8%)	8 (7.0%)	17 (14.9%)	0.121
Death, *n* (%)	12 (10.6%)	10 (8.8%)	31 (27.2%)	0.001

Abbreviations: BMI: body mass index; eGFR: estimated glomerular filtration rate; IQR: interquartile range; IU: international units; SD: standard deviation; OPN: osteopontin, MV: mechanical ventilation; RRT: renal replacement therapy.

**Table 4 jcm-10-03907-t004:** OPN and clinical characteristics independently associated with in-hospital outcomes of COVID-19 patients.

	Death (*n* = 53)				Need for Mechanical Ventilation (*n* = 104)				Need for Renal Replacement Therapy (*n* = 35)				Composite Endpoint (*n* = 192)			
	*p*	OR	95%CI		*p*	OR	95%CI		*p*	OR	95%CI		*p*	OR	95%CI	
Age (10 year increase)	0.001	1.710	1.238	2.360	0.489	0.924	0.738	1.156	0.357	0.850	0.601	1.201	0.393	1.106	0.878	1.393
Male sex	0.186	1.680	0.778	3.650	0.163	1.50	0.848	2.647	0.002	5.444	1.830	16.18	0.339	1.328	0.743	2.371
BMI (per 5 units)	0.147	1.221	0.933	1.598	0.018	1.259	1.040	1.525	0.874	1.024	0.765	1.371	0.918	0.967	0.514	1.821
Blacks (compared to non-blacks)	0.001	0.184	0.066	0.507	0.459	1.260	0.683	2.323	0.406	1.499	0.577	3.893	0.054	1.218	0.997	1.488
Current or former smoker	0.034	2.479	1.069	5.749	0.078	1.757	0.938	3.291	0.982	0.988	0.357	2.733	0.018	0.941	0.894	0.990
Diabetes mellitus	0.023	2.455	1.133	5.321	0.047	1.817	1.008	3.275	0.749	0.862	0.348	2.138	0.004	1.111	1.034	1.193
Hypertension	0.476	0.724	0.297	1.763	0.573	1.213	0.621	2.370	0.493	1.519	0.460	5.017	0.087	1.011	0.998	1.024
Coronary artery disease	0.063	0.365	0.126	1.054	0.118	0.495	0.205	1.195	0.426	0.612	0.182	2.053	0.012	2.154	1.187	3.910
Congestive heart failure	0.175	1.972	0.740	5.258	0.377	0.685	0.295	1.587	0.541	0.707	0.232	2.154	0.677	1.158	0.580	2.314
eGFR	0.080	0.941	0.879	1.007	0.105	0.959	0.911	1.009	<0.001	0.794	0.729	0.866	0.085	0.480	0.208	1.107
OPN < 55.81 ng/mL	Reference															
OPN 55.81–140.65 ng/mL	0.357	0.623	0.227	1.706	0.112	1.775	0.874	3.603	0.582	0.730	0.238	2.240	0.626	1.190	0,592	2.393
OPN > 140.66 ng/mL	0.006	3.450	1.439	8.273	<0.001	4.931	2.480	9.804	0.89	0.930	0.329	2.626	0.001	3.232	1.619	6.450
OPN (per 100% increase)	0.008	1.415	1.095	1.827	<0.001	1.598	1.308	1.950	0.639	1.077	0.789	1.470				

BMI: body mass index, eGFR: estimated glomerular filtration rate, OPN: osteopontin, OR: Odds ratio, *p*: *p*-value.

**Table 5 jcm-10-03907-t005:** Cox regression analysis for the prediction of the combined endpoint (either death, need for mechanical ventilation, or need for renal replacement therapy).

	HR, 95%CI	*p*-Value
OPN < 55.81 ng/mL	Reference	
OPN 55.81–140.65 ng/mL	1.170 (0.669–2.046)	0.582
OPN > 140.65 ng/mL	1.834 (1.071–3.139)	0.027
Age	0.996 (0.980–1.012)	0.629
Male Sex	0.902 (0.609–1.337)	0.607
Black Race	0.965 (0.625–1.488)	0.871
BMI	1.029 (1.002–1.057)	0.034
eGFR	0.992 (0.985–0.998)	0.014
Diabetes mellitus	1.526 (1.014–2.298)	0.043
Hypertension	1.204 (0.728–1.992)	0.470
Coronary artery disease	0.807 (0.454–1.435)	0.466
Congestive heart failure	1.256 (0.752–2.096)	0.383
CRP	1.312 (1.145–1.504)	<0.001

BMI: body mass index, CRP: C-reactive protein, eGFR: estimated glomerular filtration rate, OPN: osteopontin.

## Data Availability

Data from ISIC can be made available upon reasonable request through a collaborative process. Please contact: penegonz@med.umich.edu for additional information.

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
