# Peer review of "Circulating Osteopontin Levels and Outcomes in Patients Hospitalized for COVID-19"

_jcm, 2021, doi:10.3390/jcm10173907_

Round 1

Reviewer 1 Report

This manuscript reports a comprehensive statistical evaluation of serum osteopontin (OPN) levels as a potential predictive biomarker of outcome in hospitalised COVID-19 patients. Urgent assessment is required given the rapid course of disease. The study exploits ISIC's bank of documented samples looking for correlates with (i) in-hospital death (ii) mechanical ventilation (iii) need for renal dialysis. 341 patients samples are compared to healthy controls - 40 blood donors (I don't see if these were aged matched and suspect not).   

The study methods are well described and results well presented. 

MINOR COMMENTS

  • LINE 164:
    • whilst significant correlation with some other routine biomarkers is provided, I do not see data nor mention of IL-6. This warrants inclusion / comment given IL-6's known correlations with C-19.
  • TABLE 1:
    • this needs to be split into two. The comparator of candidate predictive biomarkers are lost - MAJOR please use a new Table - the comparator data is central to the purpose of the study. 
  • DISCUSSION:
    • need to include IL-6 given its recognised role in C19. 
    • Line 286 Reference re auto-antibodies to OPN does not add to understanding the case and might be omitted though to line 292. As is, is confusing.
  • FIGURES - MAJOR COMMENT
    • Each figure needs to be supported by a Figure Legend explaining the presented data (ie to be self-sufficient). Currently only a Figure Title is provided. 

Author Response

Reviewer #1: This manuscript reports a comprehensive statistical evaluation of serum osteopontin (OPN) levels as a potential predictive biomarker of outcome in hospitalised COVID-19 patients. Urgent assessment is required given the rapid course of disease. The study exploits ISIC's bank of documented samples looking for correlates with (i) in-hospital death (ii) mechanical ventilation (iii) need for renal dialysis. 341 patients samples are compared to healthy controls - 40 blood donors (I don't see if these were aged matched and suspect not). The study methods are well described and results well presented. We would like to thank this reviewer for her/his positive evaluation of our manuscript and his statement that the “methods are well described and results well presented”. By addressing all remaining issues posed by this referee, we believe that our manuscript has now gained in quality and focus, and we hope that it will be deemed appropriate for publication in JCM in its revised form. 1) LINE 164: whilst significant correlation with some other routine biomarkers is provided, I do not see data nor mention of IL-6. This warrants inclusion / comment given IL-6's known correlations with C-19. We would like to thank this referee for her/his valuable statement on a potential correlation between IL-6 and OPN serum levels. This would indeed be a very relevant analysis and we fully agree with this reviewer’s comment that a correlation between IL-6 serum levels and the severity of Covid-19 has been established previously. As our study included patients from four different international study sites, measurements of IL-6 serum levels were not performed routinely at all study centers. We have now reassessed the available data and found IL-6 serum levels for a share of 231 Covid-19 patients. In this subgroup of the study cohort, IL-6 and OPN showed a weak non-significant positive correlation of rS: 0.095 (p=0.151). Because we are unable to provide IL-6 serum levels for the whole study cohort, we did not include this novel data in the revised result section of our manuscript. However, we are happy to do so if the reviewer or the editor whishes us to do so. 2) TABLE 1: this needs to be split into two. The comparator of candidate predictive biomarkers are lost - MAJOR please use a new Table - the comparator data is central to the purpose of the study. We agree with this reviewer’s comment that table 1 was not really clear to read in its previous form. According to this reviewer’s comment, we have now split the table into table 1 and table 3. 3) DISCUSSION: need to include IL-6 given its recognised role in C19. Line 286 Reference re auto-antibodies to OPN does not add to understanding the case and might be omitted though to line 292. As is, is confusing. Both points raised by the reviewer have been addressed in the revised discussion section of our manuscript. In particular, we have addressed the predictive as well as potential therapeutic relevance of IL-6/IL-6R in the context of COVID-19 as follows: “In addition, interleukin-6 (IL-6) was suggested as a biomarker for more severe courses of COVID-19 and tocilizumab, a humanized monoclonal antibody targeting the IL-6 receptor, has been evaluated as a potential therapeutic option for COVID-19 patients and was shown to improve outcomes e.g., in critically ill patients with COVID-19 receiving organ support in ICUs.” (page 18 of the revised manuscript) 4) FIGURES - MAJOR COMMENT: Each figure needs to be supported by a Figure Legend explaining the presented data (ie to be self-sufficient). Currently only a Figure Title is provided. We have added the respective figure legends according to this reviewer’s suggestion.

Reviewer 2 Report

The paper by Hayek described the serum OPN levels in COVID-19 patients and asked the levels are associated with death or mechanical ventilation. They found that OPN levels ≥140.66 ng/ml (third tertile) increased death and mechanical ventilation but not renal replacement therapy.

These findings confirmed that the levels of OPN is a severity marker of COVID-19 infection.

Major

They should describe how tertiles are determined using the appropriate figures.

It is known there are two kinds of ELISA that measure full-length of OPN and a mixture of full-length and cleaved form. It is better to describe which is measured using the employed ELISA.

They compared various factors with OPN but only CRP was used as laboratory markers. It is better to include CRP in Table 4.

 Minor

The table should appear in order.

Author Response

Dear reviewers, please check the response to your comments in the attached document.

Round 2

Reviewer 2 Report

The revised paper was improved, significantly.

Regarding ELISA, I learned that (#BMS2066, Thermo Fischer Scientific BMS) use a monoclonal antibody that detect somewhere aa17~aa300 and the second antibody is polyclonal antibody.  This is very similar to R$D elisa which was claimed to detect both full length and cleaved form as described in https://www.mdpi.com/2227-9059/9/8/1006/pdf.   

Author Response

We would like to thank this referee for her/his careful evaluation of our manuscript. We have re-contacted our technician at "Thermo Fischer Scientific" and he confirmed that the ELISA detects both cleaved form and full-length OPN.

We have now changed the respective section accordingly:

"Circulating OPN concentrations were measured using a commercially available enzyme-linked immunosorbent assay (ELISA) that detect both full-length and cleaved human OPN according to the manufacturer’s instructions (#BMS2066, Thermo Fischer Scientific, Waltham, MA, USA)." (page 6 of the revised manuscript)

Round 3

Reviewer 2 Report

The revised paper improved.